# Etiopathogenesis of Canine Cruciate Ligament Disease: A Scoping Review

**DOI:** 10.3390/ani13020187

**Published:** 2023-01-04

**Authors:** Gert W. Niebauer, Brunella Restucci

**Affiliations:** Department of Veterinary Medicine and Animal Production, University Federico II, 80138 Naples, Italy

**Keywords:** cruciate ligament rupture, cruciate ligament disease, dog, pathogenesis, scoping review

## Abstract

**Simple Summary:**

The spontaneous rupture of the cranial cruciate ligament in the stifle joints of dogs is one of the most common veterinary orthopedic problems. Largely unknown mechanisms progressively weaken intra-articular structures which eventually fail; joint instability, osteoarthritic changes, pain, and dysfunction are the sequels. In general, surgical treatment is recommended aiming at stabilizing the hypermobile joint by a variety of surgical methods. Despite much progress in rendering surgical treatment more efficient, osteoarthritic changes, although effectively mitigated by surgery, continue, and persist. Improved knowledge on the causes of joint and ligament degradation would aid prevention and treatment. This review focuses on papers contributing to knowledge of the causes, that is, on local and systemic features, and on articular inflammatory and degenerative changes. Based on recent work, a systemic, metabolic multifactorial disease background emerged, and a new, generally accepted term has been coined: “canine cruciate ligament disease”. Primary osteoarthritis and collagen degradation seem to be the underlying key features of cruciate ligament disease. Besides redefining the pathogenesis in the dog, these findings render the canine joint disease a potentially useful clinical animal model for human osteoarthritic diseases. Thus, trying to unravel the enigma of spontaneous cruciate ligament disease may benefit the treatment of both canine and human degenerative joint disease, in general.

**Abstract:**

The spontaneous rupture of the cranial cruciate ligament in dogs remains a pathoetiologic puzzle. Despite much progress in research over the past years, the systemic and local mechanisms leading to ligament degeneration and structural failure remain largely obscure. This scoping review focuses on pathogenesis and aims at summarizing and interpreting today’s knowledge on causes of canine cruciate ligament rupture, i.e., the multifactorial mechanisms leading to degenerative stifle joint disease with collagen matrix degeneration and structural failures. Thus, the initial view of traumatic ligament rupture, fostered by “wear and tear”, has clearly been replaced by a new concept of systemic processes linked to progressive degenerative joint disease and ligament failure; thus, the term “cranial cruciate ligament disease” has been coined and is generally accepted. In addition, cruciate ligament rupture in people shares some similarities with the lesion in dogs; therefore, the review also includes comparative studies. The methods used were based on the PRISMA-ScR model (Preferred Reporting Items for Systematic Reviews and Meta-Analyses Extension for Scoping Reviews).

## 1. Introduction

Spontaneous cranial cruciate ligament rupture (CCLR) in dogs is one of the most frequently seen conditions in veterinary orthopedics and carries the highest economic impact in orthopedic patient care (in the US) [1]. This painful and debilitating joint lesion is commonly treated by surgically reducing hypermobility, which ensues after ligament failure. Despite the high overall incidence (~2.55%) with a tendency to increase [2], pathoetiology remains obscure and treatments are largely symptomatic and address biomechanics. There is now consensus that CCLR has a multifactorial background involving local and systemic mechanisms, with osteoarthritis (OA) being a key feature. Unraveling the pathomechanisms of this inflammatory and degenerative joint illness is challenging; this is also true because of its biphasic evolution: during a clinically nearly silent initial phase, progressive collagen matrix degradation of the cruciate ligament develops and persists [3,4]. This leads eventually to structural failure, most often in the mid-section and without excessive load [5,6]. Ensuing joint instability exacerbates inflammatory and degenerative changes in a second phase (secondary OA) [7]. Progression of secondary OA can be slowed by joint stabilizing surgery; however, the primary, underlying osteoarthritic disease process continues and prevents, in most cases, the full return to integrity [8,9,10,11,12]. Thus, the initial pathogenic concept of “wear and tear” followed by ligament rupture [13] has clearly changed to a new understanding of the joint as an organ affected by complex and largely idiopathic disease mechanisms leading (among other changes) to cruciate ligament failure [14]; time-delayed, in more than 50% of cases the contralateral stifle joint is similarly affected [15,16,17]. Based on these newer insights, the term “Cranial Cruciate Ligament Disease” (CCLD) has been coined [7], and similarities between the anterior cruciate ligament ruptures (ACLR) in people became increasingly apparent: that is, in humans as in dogs, spontaneous (non-traumatic) cruciate ligament tears are now defined as “non-contact” injuries with unclear etiology [18]. Thus, in both species, and despite different conformational biomechanics, idiopathic degenerative mechanisms weaken ligaments and cause failure, clearly unrelated to a single traumatic event [19]. This scoping review aims at identifying and interpreting studies that contribute to pathogenesis and to disease mechanisms in canine CCLD; in addition, papers addressing the comparative pathoetiology of cruciate ligament rupture are included. Although several review articles focusing on possible causes of CCLR were published previously [14,20,21,22,23,24,25], an update in the form of a scoping review centred on etiopathogenesis may be warranted. New directions for future studies may become identifiable, with the aim to improve the treatment of spontaneous CCLR through a better understanding of disease mechanisms.

## 2. Materials and Methods

The scoping review was based on the PRISMA-ScR (Preferred Reporting Items for Systematic Reviews and Meta-Analyses Extension for Scoping Reviews) model. Publications to be evaluated were extracted by the following search criteria in PubMed, Web of Science, and in the author’s personal database: “canine cruciate ligament rupture OR cruciate ligament disease AND pathogenesis OR (a)etiology”. In PubMed and Web of Science, on 17/07/22, this search yielded 271 hits, of which 141 were eliminated by the following exclusion criteria: studies on experimental models (Pond-Nuki), on surgical treatment and complications thereof, and on diagnostic imaging, when unrelated to pathogenesis. To the remaining 130 papers, several older texts were added, which were retrieved by the above search criteria in the author’s reference collection. Some of these add-ons are in languages other than English and not listed in electronic databases because of the early year of publication. For enhanced comprehension, a short introductory section reviewed the anatomical, physiological, and biomechanical features of the stifle joint, focusing on descriptions with links to pathophysiology. Then, and based on their focus on pathoetiology, papers were grouped as follows (with occasional overlap):

Cruciate ligament anatomy, physiology, biomechanical features (Section 3.1).

Risk factors: breed, sex, neuter status, weight, age, activity (Section 3.2).

Genetics (Section 3.3).

Biomechanics/joint functional anatomy/orthopedic conformation (Section 3.4).

Osteoarthritic changes (Section 3.5).

Inflammation, cytokines, immune mediation, apoptosis (Section 3.5.1).

Synovial membrane, matrix collagen, ligaments, menisci (Section 3.5.2).

Systemic factors (hormones, metabolites, diseases, infections, immune system) (Section 3.5.3).

Late-stage osteoarthritis (Section 3.5.4).

## 3. Results

### 3.1. Cruciate Ligament Anatomy, Physiology, Biomechanical Features

The functional anatomy of the ligaments and menisci of the stifle joint, aiming at explaining failure and evaluating joint stabilization techniques, was extensively studied by Arnoczky et al. and more recently by de Rooster et al. [26,27,28,29,30,31]. The cranial cruciate ligament (CCL) was found to have a multifascicular structure consisting of multiple collagen bundles, whose spatial orientation is directly related to its function as a constraint of joint motion. This arrangement results in a different portion of the ligament being taut and therefore functional, throughout the range of motion. The metabolism of the CCL is provided through its intrinsic blood supply and by diffusion through an enveloping synovial sheet [32]. The synovial envelope, when intact, shields the CCL from direct synovial fluid contact, so that the CCL, at least functionally, can be considered an “extra-articular” structure. Periligamentous vessels from this synovial envelope penetrate the ligament transversely and anastomose with a longitudinal network of endoligamentous vessels which originate from the proximal and distal osteochondral insertions. However, those femoro-tibial attachments do not contribute significantly to the vascularity of the CCL [33]. Specifically, the central aspect is poorly vascularized and features endarterial loops [34]; this is also the zone of initial ligament degeneration and rupture [6,35,36,37].

For recent reviews on biomechanics of the stifle joint as related to CCLD etiopathogenesis, see Cook and Spinella et al. [14,23]. Among the many forces acting during locomotion on the cruciate ligament, cranial tibial trust seems to exhibit the strongest load which is counteracted by the CCL [22]. This cranially directed shear force is suggested to increase during weight-bearing dynamically with increasing caudo-distal inclination of the tibial plateau [38]. Neutralizing tibial trust has become the gold standard of surgical therapy; this is achieved by either lowering the inclination angle of the tibia plateau through specific rotational osteotomies (TPLO) [39,40], CORA-based leveling osteotomy (CBLO) [41], cranial closing wedge osteotomy (CCWO) [42], or by osteomyzing and advancing the tibial tuberosity to achieve a dynamically neutralizing angle of ~90° between the patellar ligament and tibial plateau (TTA) [43,44].

### 3.2. Risk Factors: Breed, Sex, Neuter Status, Weight, Age, Activity

A steep tibial plateau angle (>30°, depending on dog size and breed) [22,45] and narrow relative width of the tibial tuberosity [46] are conformational risk factors which are aggravated by higher-than-normal body weight [47]. Early neutering (less than 12 months of age) is a risk factor for developing an increased tibial plateau angle and, in general, increases by 5% in males and by 8% in females the risk for CCLR [45,48]. Contralateral CCLR may occur time-delayed between 1 and 2 years in ~40–50% of cases [49]; large breed dogs and the severity of osteoarthritic changes are factors of increased risk also for bilateral CCLD [16,17,50], while athletic body conformation in agility dogs, on the contrary, decreases the risk of CCLR [51]. Breed-related risk is generally linked to heavy body conformation and large breed dogs; for a listing of breeds, see [2]. Some breeds, however, have a notably low prevalence or quasi absence of CCLD, e.g., Dachshund, Greyhound, Afghan, Shi-tzu, and Pekingese. Although speculative as for a causal link, it may be of interest that in Dachshunds the CCL seems to have a denser vascular network throughout the ligament as compared to other breeds [52]. Intact female dogs, overall, are twice as likely to develop CCLD as compared to males [53]; neutering in both sexes increases the risk of CCLR [54], although obesity, linked to neutering, might be a cofounding factor. Obesity per se quadruples the risk of CCLR [53]. Heavy body weight, whether as a normal feature of large breed dogs or linked to obesity, increases the risk of ligament failure [47,55,56]. Age, as in most degenerative joint diseases, is a risk factor: regardless of breed, this risk peaks around 8 years of age; however, breeds with a high prevalence for CCLD (e.g., Boxer, Labrador Retriever, Rottweiler), tend to develop the disease earlier in life [47,57]. Physical exercise and athletic constitution, on the other hand, are negative risk factors [51]. For epidemiological studies on larger cohorts, see also [55,58].

### 3.3. Genetics

There is a strong breed-related predisposition for CCLD [47,59,60,61,62,63]; recently, several genes (single nucleotide polymorphisms) common to dogs with a high risk for CCLR were identified through genotyping [64,65]. The set of key genes identified in susceptible dog breeds is coding for collagen strength and stability and is involved in extracellular matrix formation [66,67]. Thus, on the morphologic level, a genetic background has been identified concerning ligament formation and strength [68,69]. On the other hand, in another study, no difference in gene expression has been found [70]. However, candidate genes were identified to be involved in tibial plateau slope formation and in developing a compressed infrapatellar fat pad, i.e., a surrogate for stifle osteoarthritis and CCLR [71]. 

### 3.4. Biomechanics/Joint Functional Anatomy/Orthopedic Conformation

The healthy CCL is among the strongest ligaments [72], with an average tensile strength of 92 N/mm^2^ or 18.2 Megapascals [73]. External breaking force causes an avulsion fracture, especially in younger dogs, rather than a failure of the ligament itself. On the other hand, and in the majority of spontaneously ruptured CCLs, tearing occurs in the central part of the ligament and is preceded by degenerative processes and collagen degradation [7,13]. This central section of the CCL is poorly vascularized [32,34,52,74], giving rise to the concept that the vascular microenvironment of this core region is an underlying condition for ligament failure [36]. Degenerative changes, immune complex deposition, lack of scar formation, and insufficiency of healing all may be linked to the peculiar microvascular anatomy of the CCL [6,75,76,77]. The following studies focused on biomechanics of the stifle with CCLR and are based on the concept of the joint being an organ [78], which includes intercondylar notch [79], cartilage [80,81,82], joint capsule [83,84], synovial fluid, menisci, collateral ligaments [14,23,27,28,30,31,80,81,85,86,87,88,89,90,91,92,93,94], the patella [95], and articulating bones [96]. 

### 3.5. Osteoarthritic Changes 

Osteoarthritis (OA) is a key feature of CCLD [21,97]. Early osteoarthritic changes are already identifiable in stifle joints with little or without instability, such as in cases of partial rupture [3,98,99,100,101,102]. The use of advanced imaging techniques and arthroscopy clearly demonstrate the presence of inflammatory and degenerative changes prior to ligament failure and joint instability [92,103,104,105]. This largely idiopathic phase of primary OA is the subject of most studies in search for etiopathology; in contrast to primary OA, secondary OA can be studied in animal models (Pond-Nuki model), such as by the experimental transection of the cranial cruciate ligament [106,107]. In spontaneous CCLR, joint degrading processes precede instability, are only enhanced after rupture, and are ongoing [108]. Thus, and despite surgical joint stabilization, OA progresses, although it is mitigated in relation to the efficacy of the stabilization method [8,109,110,111,112,113,114,115,116]. Lesions of menisci and other intra-articular structures often accompany ligament failure or are the sequel of instability; they may contribute to and enhance OA [9,80,81,87,88,89,90,91,92,94,103,117,117,118,119]. Non-surgical treatment strategies aim therefore to identify early-stage OA; to that end, the search for biomarkers of OA and for metabolites of structural changes is ongoing in both human and veterinary medicine [108,120,121,122]. 

#### 3.5.1. Inflammation, Cytokines, Immune Mediation, Apoptosis

Collagen degradation is a key feature of OA, mediated at large by matrix metalloproteinases (MMPs), mainly MMP-1,2 and 13 [123,124], as well as MMP-3 [125,126]. Matrix collagenase activity in ruptured canine cruciate ligaments samples has been described early on [127,128] and has been linked with collagen type I fiber creep and ligament failure [129]. On the other hand, collagenase-generated collagen type II metabolites, deriving from joint cartilage, have not been found elevated in CCLR [130]. In synovial fluid samples of dogs with CCLR and OA, the upregulation of degradative enzymes, metabolites, and inflammatory cytokines has been demonstrated in several studies: IL-1β, IL-6, IL8, and TnF-α expressions correlate with inflammatory cycles of OA [131,132] and decrease after successful surgical treatment [133]; however, the search for cartilage-derived OA biomarkers such as fibronectin [134] and keratan- and chondroitin-sulfate epitopes has not yielded clinical usefulness so far [125]. While cartilage-derived nitric oxide metabolites were found increased in canine OA, no significant correlation with CCLR was found [135]. However, nitric oxide seems to mediate cell death and apoptosis of ligamentocytes in CCLD, expressing stronger effects in cranial cruciate ligaments as compared to the caudal CL [136], the collateral ligaments of the stifle, and the round ligament of the femoral head [137]. That programmed cell death may play a role in CCL degeneration found confirmation in a later study by the same group [138], yet it remains open as to whether apoptosis is an epiphenomenon or an etiologic factor. The fact that apoptosis has equally been found in partially ruptured CCL demonstrates that apoptosis is already present in the early stages of CCLD [101]. However, other trigger mechanisms besides nitric oxide seem involved in programmed ligamentocytes death, as the selective blocking of nitric oxide does not influence apoptosis [139]; however, mediators of apoptosis cause more fibrocyte death in cells derived from cranial cruciate ligaments as compared to those of the caudal ligament [140]. Among inhibitors of nitric oxide, doxycycline has been investigated as a potential medical treatment for OA and CCLR [141]; its efficacy to reduce nitric oxide via reduced stromelysin production has been shown in OA cartilage samples but not in ruptured cruciate ligaments [142]. Similarly, doxycycline, a presumptive downregulator of intra-articular MMP activity, has been shown to be relatively ineffective to reduce inflammatory changes in CCLD postoperatively [124]. A recent systematic review evaluating the efficacy of doxycycline to treat CCLR-related OA confirmed these mixed results, but also stated that, clinically, some positive results in terms of decreased inflammation and pain were seen [143]. 

#### 3.5.2. Synovial Membrane, Matrix Collagen, Ligaments, Menisci

Inflammatory changes, typically seen in CCLD, affect the entire joint, especially the synovial membrane. Synovitis is already present prior to ligament failure, demonstrating an early and ongoing inflammatory collagen degrading process [5]. In later stages and in unstable joints, the inspissated and often villous synovial membrane contains largely mononuclear cells. Abundant B- and T-cell lymphocytes, plasma cells, tartrate-resistant acid phosphatase (TRAP)-activated macrophages, and dendritic cell infiltrates are present [144,145,146,147]; the degree of these inflammatory changes is linked to the degree of degenerative changes within the CCL [62]. Although bacterial DNA has been occasionally detected by PCR in cases of CCLR [148], an infectious etiologic component of CCLD is unlikely. Nevertheless, the type and quantity of synovial cellular infiltrates are signs of an immune response; immune-mediation as one possible pathogenic factor was first demonstrated by finding C1q-binding immune complexes within ruptured CCLs and joint capsules [76,149], and a link between intra-ligamentous immune complex deposition and collagen type I fiber degradation could then be demonstrated [129]. It has been suggested that the trapping and deposition of immune complexes may be enhanced by the particular and scarce blood supply of the central part of the CCL, containing endarterial vessels [31,32,37]. The ensuing hypothesis that the IgG of the immune complexes might have epitopes against ligament-derived collagen type I was later-on confirmed [150,151,152]. These antibodies were found in synovial fluids of affected stifles and to a lesser degree in the contralateral stifles, and in circulation, indicating systemic immune reactions [153]. The simultaneous bilateral CCLR is relatively rare, but time-delayed it is relative common; one study described contralateral CCLR as occurring in one-third of the dogs within a year after the first rupture [57], others reported a somewhat higher incidence of ~50% with mean intervals of ~2.5 years [15]. Osteoarthritic changes (osteophytes), when identified bilaterally in dogs with unilateral CCLR, are considered risk factors for insipient contralateral ligament failure [154].

A fair number of studies focused on degenerative changes within the ligaments. The following morphologic/anatomical factors have been identified as risk factors for mechanic and/or metabolic damage to the CCL [85]: tibial plateau angle and tibial trust [155], the above-stated poor intrinsic blood supply [33,52,156] (Figure 1), and narrow femoral intercondylar notches, among others, have been cited [79,157,158].

Micromorphologic studies of completely or partially ruptured ligaments showed the following: ligamentocyte (fibroblast) transformation into spheroid cells, fibroblast necrosis, and fibrocartilaginous metaplasia are the principal cellular changes [6,75]; most authors attributed these cellular transformations to hypoxic metabolism in the poorly vascularized core of the CCL. Apoptosis seems not to play a role as a promotor of fibroblast decay [101,140]. Macrophages are the predominant extrinsic cells and are scavengers of C1q-binding immune complexes [149]; as antigen-presenting cells, they play a key role in immune responses and in the cytokine-induced upregulation of proteolytic enzymes (collagenases) [159]. Thus, lymphocytes [160] and TRAP+ macrophage-like cells migrating from the epiligament have been implicated as promotors of progressive CCL degradation [161,162]. In addition, the increased expression of immune-response genes for cathepsin K, MMP-9, and TRAP was found in the synovial fluid of dogs with CCLD [152,161].

#### 3.5.3. Systemic Factors

Based on the results of these studies, a multilevel hypothesis of immune-mediated pathomechanisms has emerged. Doom et al. have published a well-illustrated comprehensive overview of all possible implications of humoral and cell-mediated immune responses and their interactions in CCLD [163]. In short: the intact CCL is shielded from the joint space by a thin synovial membranous envelope (epiligament); when this membrane ruptures early on in the joint disease, degenerated collagen type I fibrils become exposed to synovial fluid and may evoke an (auto-) immune response; circulating anti-collagen antibodies may contribute to contralateral CCLD and local cellular responses trigger cytokine cascades and proteolytic collagen matrix degeneration; the destructive processes seem to progress in the form of a vicious cycle, whereby enhanced exposure to collagen metabolites enhances immune-mediated inflammatory reactions, which in turn upregulate proteolysis, augmenting exposure to collagen-derived epitopes [164]. These pathomechanisms might be enhanced or accompanied by immune complexes trapped in the end-arterial loops-containing microvasculature of the central part of the CCL [149]; the macrophage and dendritic cell-derived antigen presentation might therefore not only originate from scavenging cells migrating from the epiligament, but also from intra-ligamentous macrophages. 

Interestingly, in rheumatoid arthritis of dogs, bilateral cruciate ligament disease often develops with similar pathologic features [164].

There is agreement on collagen lysis as being a key factor in cruciate ligament degeneration; however, the initiating mechanisms triggering ligament fiber decay remain unclear [165]. It has long been suspected that the gender and hormones of reproduction may be linked to CCLR [166]; in fact, intact female dogs (independent of other risk factors) are twice as likely to develop CCLD as compared to intact male dogs [53]. In a large cohort of over 3000 dogs, the overall prevalence of CCLR was 3.48%; gonadectomy increases prevalence in both sexes and was found as 2.09% in intact males and 5.15 % in neutered females [167]. These epidemiologic features, linked to the previously stated molecular biologic data, are in support of hormone-related pathomechanisms: for instance, female sex hormones have been found to upregulate MMP-mediated collagen degradation [168]. As described in human medicine, women in general, and female athletes in specific [169], inherit an increased risk for ACL [19,170]. In women, knee joint laxity and ACL have been found linked to the reproductive cycle, hand-in-hand with surges of estrogen and relaxin [171,172]. Relaxin modifies and weakens the molecular structure of collagen, causing fiber sliding, creep, and joint laxity [173]. Relaxin involvement in cruciate ligament rupture has been demonstrated in women [174] and in dogs of both sexes [175]. These latter insights are in support of the general concept of a stifle joint disease with the involvement of systemic/humoral factors.

#### 3.5.4. Late-Stage Osteoarthritis

In researching pathogenic mechanisms, most of the reviewed studies focused on the early stages of the joint disease (primary OA). Nevertheless, late and end-stages of CCLD should also be reviewed: secondary OA invariably progresses, especially in untreated cases, accompanied by typical chronic inflammatory changes (osteophytes, cartilage damage) [104,176]. The severity of degradation largely depends on the cited risk factors: heavy body weight, breed, age, sex, and neuter status. However, independent of these factors, in all dogs with CCLR, the fate of the ruptured ligament is the same: healing and neo-vascularization do not take place, and within 2 to 4 months after ligament rupture, collagen fibrils will have been lysed and absorbed, or reduced to stump-like remnants, lined with a thin inflammatory membrane [177,178], such as that exemplified in Figure 2. 

Osteophytes formation (Figure 3) is another feature of severe chronic OA together with joint capsule fibrosis [115,116]. These changes enhance pain and disuse on one hand, but on the other, proliferative remodeling contributes to joint stabilization. In fact, in such advanced stages of OA with surgically untreated CCLR, on clinical examination, the typical drawer sign (cranio-caudal sliding of articulating bones) tends to disappear and becomes very subtle. In end-stage ankylosing OA, although debilitating, the decreased range of motion may result in decreased ambulatory pain. Thus, when seen from a pathophysiologic standpoint, it may even be argued that end-stage OA is the result of a (failed) attempt to “heal” an internally deranged joint through lysis of debris, by osteophyte formation, and joint capsule fibrosis, resulting in a functionally impaired but increasingly stable joint. 

## 4. Discussion

Although a clear understanding of the etiopathogenesis of CCLD is still lacking, several noteworthy contributions, helpful in unravelling the multifactorial causes, have been made over the last decade. There is now strong evidence that ligament failure is preceded by a clinically relative silent and progressive phase of collagen matrix degeneration which structurally weakens intra-articular structures, and above all, the CCL. Inflammatory changes, characterized by largely mononuclear chronic synovitis, progressively affect the entire joint [62]. Pathogenetic studies on a larger scale have been hampered in the past by the spontaneous nature of the disease, which cannot entirely be replicated by experimental transection of the CCL (Pond-Nuki Model) [179]. Therefore, samples for tissue-based research in client-owned dogs during the early disease stages through surgical biopsy could only occasionally be obtained. Today, however, the wide use of mini-invasive arthroscopy and late-generation diagnostic images do provide new information on subtle changes; thus, villous synovitis, partial ligament tears, collagen fibrillation, cartilage, and occasionally meniscal damage become detectable in the earlier phases of the disease, when joints are biomechanically still intact [4]. Through direct endoscopic visualization, it has also become evident that the caudal cruciate ligaments undergo similar degenerative changes, although they rarely rupture [104]. These and the finding that the contralateral stifle joints are commonly affected by the same disease process, either simultaneously or time-delayed, are additional support for the generally accepted concept of seeing the synovial joint as an organ, and consequently CCLR as being an organ disease with systemic features. 

On the clinical level, progressive ligament weakening (partial ligament tear), together with increasing inflammatory changes, cause joint laxity and pain. Eventually, CCL failure ensues, resulting in sudden joint instability, in intra-articular debris, and exacerbated inflammatory reactions (secondary OA). Collagenolytic mechanisms pursue, and CCL remnants, when not surgically removed, are slowly metabolized by proteolysis and phagocytosis; metabolites enhance inflammatory/degenerative changes, which in untreated cases may result in sustained, severe OA [116]. Factors such as heavy body weight and breed disposition negatively affect the outcome. However, dogs with less than 15 kg body weight, treated non-surgically, have an about 75% chance of return to full joint function, albeit with progressing OA [24,180]. These data have not been updated since on a larger cohort; a re-evaluation seems useful in the light of today’s general trend to surgically treat CCLD, even in small, light-weight dogs [40]. In untreated cases, it is however noteworthy that chronic inflammatory changes may, in the end, reduce instability through joint capsule fibrosis and osteophytosis. This stabilizing effect, however, is in general not outweighing the negative effects of the degrading inflammatory mechanisms, especially in heavy-weight subjects [106]. 

The following key questions remain still incompletely answered: (1) which mechanisms initially trigger ligament collagen degeneration? (2) Why does progression to structural failure occur ? Based on the here reviewed findings and in synthesis, the following chain of biomechanical and biomolecular events may underly CCLD:

The intact CCL prevents caudo-cranial translation of the tibia; therefore, a steep caudo-distal slope of the tibial plateau, enhanced by a narrow femoral notch, is seen as the main biomechanical stressor for the cranial CL [157]. In addition, musculoskeletal factors such as the integrity of the quadriceps mechanism as the agonist of the CCL, play a role in translational forces. Quadriceps amyotonia may thus have a negative effect in CCL-deficient joints. Heavy body weight and/or obesity add to the strain and to laxity, if present. Although the CCL, by definition, is an intra-articular structure, the intact epiligamentous envelope effectively shields the CCL from the synovial joint space. Whether or not it is mechanically induced, laceration of this well-vascularized synovial sheet (epiligament), is one of the first observable intra-articular lesions. Disintegration of or damage to the envelope results in a reduced blood supply, and “unmasks” the ligament, its debris, and metabolites, rendering the CCL a true, poorly vascularized “intra-articular structure” [36,37]. The damaged, fibrillating collagen type I matrix becomes thereby exposed to immune-competent cells of the synovial joint capsule via direct contact with synovial fluid. Mononuclear, largely plasmacellular synovitis and the appearance of local and circulating collagen type I antibodies strongly suggest an autoimmune reaction [163]. Whether immune mediation is an etiological component or the inflammatory sequel of the intra-articular exposure of collagen debris remains unanswered at present. Nevertheless, enhancement of joint inflammation through a vicious circle of antigen (collagen type I) presentation, macrophage/dendritic cell activation, and MMP upregulation, resulting in further collagen degeneration, has been well documented [181]. 

Another contributing factor may be the relatively scarce internal blood supply of the CCL as compared to the caudal CL. This is especially the case in the mid-section, the very area where histologically the first degenerative changes are verifiable. This may lead to hypoxic matrix degradation, which per se is unlikely to be causative: it would be highly improbable that during evolution, such imperfectness had withstood Darwinian selection. Yet, the demonstrated absence of healing of ruptured CCLs may well be connected to the relative avascularity [34]. On the other hand, the ligamentous micro-vasculature, containing end-arterial loops, favors immune complex deposition. Trapped immune complexes, linked to anti-collagen antibody aggregation, have been found in ruptured CCL [76]. Immune complexes, triggering cytokine-mediated inflammatory pathways, in turn, may upregulate MMPs, causing further collagen decay; thus, a cycle of sustained immune-mediated collagenous matrix degeneration may result, enhanced, and perpetuated by intra-articular debris and ligament remnants. Several months after CCLR, when collagenous debris has been lysed and absorbed, painful inflammatory reactions diminish or subside in many patients, especially in small dogs who may regain satisfactory limb function several months after untreated CCLR, despite progressing OA [180]. This may also lead to the assumption that after CCLR, joint instability is the major factor of sustained OA in heavier dogs. Similar conclusions can be drawn from postoperative long-term observations: the slow progression of OA cannot be entirely avoided by any dynamic or static joint stabilizing technique, as none can render a cruciate deficient joint completely stable [182]. 

In searching for mechanisms other than immune-mediated inflammatory pathways upregulating MMPs, the peptide hormone relaxin has recently been implicated. Relaxin may play a role in early phases of CCLD. By binding to its cognitive cellular receptors, relaxin is a potent activator of matrix collagenases in target tissues. Such receptor binding has been found in the fibroblasts (ligamentocytes) of ruptured CCL. Collagen fiber sliding and crimping, joint laxity and partial CCLR, thus, may be linked to relaxin-induced lysis of collagen type I cross-links. If confirmed in additional studies, relaxin could be involved in the early phases of cruciate ligament weakening and decay. However, such an assumption must be taken with caution as relaxin is well-known for its connective tissue re-modeling properties [183]; therefore, relaxin, not unlike anti-collagen antibody formation, might only be an epiphenomenon of the initial collagen matrix degeneration, which nonetheless remains still idiopathic. 

Beyond the scientific insight, the primary goal of pathoetiologic studies is prevention and therapy. Surgery still is the gold standard of therapy with generally excellent results; it has been shown that long-term outcomes are significantly better when dynamic joint stabilization is effectuated in early stages of CCLD, i.e., in cases of still incomplete ligament rupture, as compared to complete ligament tear [105]. This clinical observation might well be linked to a relatively small load of intra-articular collagen debris at the time of surgery, and a consequently lower degree of immune-mediated joint inflammation, the progression of which might be reduced by joint stabilization. The general principle of early disease recognition and early treatment is thus an important consideration also in CCLD. Recognizing these early disease stages can be challenging. Despite long-term efforts in human as well as veterinary medicine, the search for early OA markers in general and in CCLD has not yielded clinically applicable results [97]; nevertheless, orthopedic examination, arthroscopy, and newer medical imaging techniques are helpful in identifying primary OA stages without yet fully developed joint instability. The responses, if any, to medical treatment target the two main known factors of CCLD: inflammation and MMP-derived collagen decay. Thus, NSAIDs and MMP-inhibitors are used with varying results, showing clinically better results in early disease stages [141].

To render non-surgical treatment more acceptable and to develop preventive measures, it seems paramount to gain insight into the initial pathomechanisms of CCLD; that is, the cellular and molecular mechanisms which likely stand at the very beginning of the chain of events and develop before the majority of the here reviewed pathomechanisms become apparent: immune-mediated inflammation and MMP dysregulation. Future research strategies shall shine a light onto these still-obscure triggering mechanisms, pivotal for a better understanding of the causes of CCLD.

## 5. Conclusions

According to the reviewed literature, the present state of knowledge on the pathoetiology of CCLD may be summarized as follows: the spontaneous structural failure of the CCL is preceded by a clinically silent phase of primary OA, during which the progressive weakening of the cruciate ligament occurs. Matrix metalloproteinases progressively decompose and lyse the CCL, enhanced by tissue hypoxemia and immune-mediated inflammatory changes. Protracted exposure to collagenous debris, acting as the antigen, stimulates the formation of synovia-bound and circulating anti-collagen autoantibodies as well as local immune complex deposition, which, in return, mediate collagen proteolysis. Weakened sufficiently by matrix decay and partial tears, the cranial CL eventually fails, while the caudal CL, affected to a lesser degree, remains functionally intact. Final structural failure of the cranial CL by non-contact injury is linked to its biomechanics, to joint conformation, to blood supply and joint metabolism, as well as to body weight and muscular envelope. Other risk factors such as genetics (breed), sex, neuter status, and obesity are involved as well. Yet, scarce information is available on the etiologic mechanisms acting on the local cellular and molecular level and likely initiating collagen decay. One candidate for an early MMP-upregulating mechanism may be relaxin-related collagen fiber degeneration, substantiated by the finding of relaxin/receptor binding on ligamentocytes. Whether this and/or the vicious cycle-like collagen degeneration by immune-mediated processes are only epiphenomena or are disease-initiating pathomechanisms, remains still unanswered.

## Figures and Tables

**Figure 1 animals-13-00187-f001:**
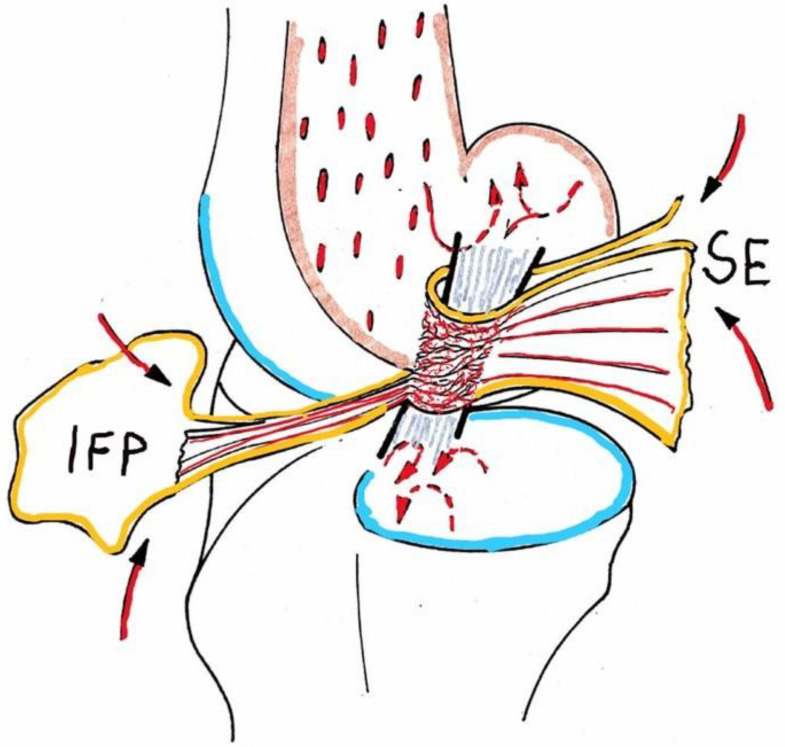
Schematic drawing of the blood supply to the canine cranial cruciate ligament; IFP, infrapatellar fat pad, SE, synovial envelope (epiligament): arrows indicate afferent supply, broken arrows show endosteal vessels, only marginally (from proximal) or not entering ligament matrix (from Niebauer GW, Pathomechanisms in canine cruciate ligament rupture <in German>. PhD Thesis 1982, Vet. Med Univ. Vienna, Austria).

**Figure 2 animals-13-00187-f002:**
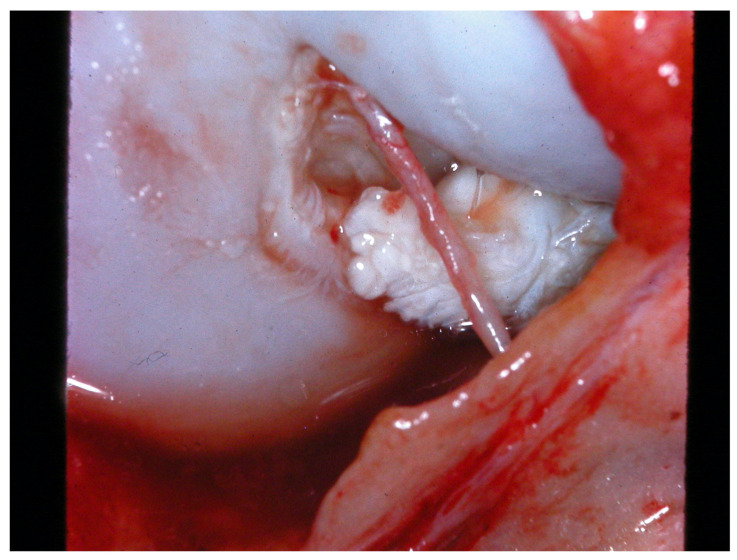
Intra-operative image of a cranial cruciate ligament ruptured 4 weeks previously; note the distal ligament stump with rounded fibrillar edges due to ongoing collagenolysis; surface is partly covered by an inflammatory tissue membrane (reddish patches). The structure visible in front of the ligament is the remnant of the epiligamentous synovial shield which covered the intact cruciate ligament (image by the author).

**Figure 3 animals-13-00187-f003:**
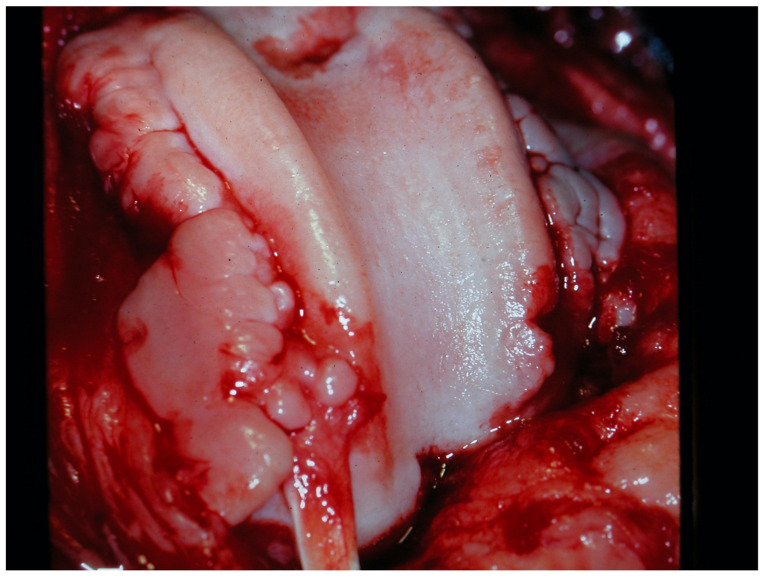
Late-stage OA in a dog 4 months after untreated CCLR; note the bilateral intra-capsular osteophytes and cartilage erosion on femoral trochlear ridges; the tendinous structure is the insertion of the extensor digitorum pedis longus muscle (image by the author).

## Data Availability

Not applicable.

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
