# Peer review of "Etiopathogenesis of Canine Cruciate Ligament Disease: A Scoping Review"

_animals, 2023, doi:10.3390/ani13020187_

Round 1

Reviewer 1 Report

The Authors performed a deepened review about the etiopathogenesis of cranial cruciate ligament rupture in dogs.

General comment

- The paper is well written, even if the topic is complex, the Authors made it easily readable.

I agree with the Authors about the term “disease”, but a strong evidence of a translational animal model to human is still lacking and from this paper does not appear. I suggest to remove or to expand the concept of "One Health" through the manuscript. 

- I suggest to revise and reduce the numbers of bibliographic references, to make them consultable by the readers.

- I suggest to correct the refences numbers through the text: e.g line 49: [3][4], must be indicated as" [3,4]".

- I suggest to check the reference style and to remove DOI numbers that is not required

- A strong evidence of surgical treatment doesn't emerge from the manuscript, even if "surgery is effective but not curative" must be expanded; because nowadays there are no better treatments than the corrective osteotomies technique.

Line comment:

- Line 44: "palliative" needs a reference, is a quite strong evaluation.

- Lines 60-61: please add a reference, and better describe the similarities that are only related to the etiopathogenesis, because biomechanics are obviously different.

- Line 118: Craniad. Please correct

- Line 121: I would add CBLO with references

- Line 130: "early neutering" can you expand this concept?

- Line 140: desexing: please change with neutering

- Line 164: "prrocesses" retype

- Line 185: reference 117 is repeated, check the reference style

- Line 222: I would remove the sentence about author's unpublished data in a review paper.

- Figure 1: the figure is hard to understand it would be better with colors.

- Line 262: double dots at the end of the sentence

- Figure 2: line 324, 4 weeks before was the rupture or the emerging of clinical signs? 

- Line 329: Ostephyte. Add "s"

- Line 333: please change "antero-posterior" with cranio-caudal

- Line 333: it is not correct that the drawer test disappears; (Leumann, 2019), "unpublished clinical data" please remove

- Line 338: please remove "autostabilisation"

- Line 372: Sentence is consequence of an old reference  (Vasseur, 1984).In this study, recovery for small breed dogs was slow and took an average of 4 months; 19% of small dogs with an apparent resolution of lameness had clinically evident muscle atrophy, 67%

had increased medial buttress, 100% had evidence of radiographic progression of OA, and 43% had cranial drawer instability.

- Lines 384-387: please add a reference to the statements.

- Lines 404-405: please add a reference

- Lines 406-408: the sentence is unclear

Author Response

Please find attached Word file with replies to reviewer 1

Reviewer 2 Report

Etiopathogenesis of canine cruciate ligament rupture: a scoping review

Thank you for your thorough overview of CCLD.

Remarks:

L2: Title: CCL disease instead of rupture. Pleased adapt

L4: 1 is missing for both authors. Please adapt

L10: In general, surgical treatment is recommended ….. Please adapt

L16:  …degenerative changes. Based on recent work…

L42:  …by surgically treating hypermobility. There is no true stabilization. Please adapt

L52: secondary OA cannot be halted. Please correct

L68: Ref 7 and 21 are a duplication. Please correct

L75: How many publication from Web of Science? Not mentioned now.

L115; CCL Rupture or Disease. Etiopathogenesis as in title and consistent throughout the manuscript. Now also as pathoetiology.

L118: shear force is suggested to increase ….. Please adapt

L126: Steep TPA. 25-30 is normal. This is confusing. Please adapt.

L147: Please summarize Ref 55, 58, 59

L150: a candidate gene is incorrect. Several genes are implicated. Please correct

L164: degenerative processes…Please correct

L172: Ref 93 and 95 are a duplication. Please correct

L185: Ref 117 duplicated in the text.

L208degeneration finds… Please remove ´is´.

L254: Ref 168 = 28. Please correct duplication

L256: Delineation of structures is missing from Figure 1.

L328: String-like structure belongs to normal anatomy of the stifle joint. Nomenclature unfortunately ambiguous

L335: decreased ambulatory pain. Please remove `de`

L341: there is only one trochlea therefore not trochleae. Otherwise use the term trochlear ridges

L447: has not yielded clinical applicable results

L511: Cook JL Biology versus Biomechanics. Please correct

L592: Ref 55 = 98. Please correct duplication

L856: Ref 168 = 28. Please correct duplication

Author Response

Thank you very much for your review with most valuable suggestions, comments and corrections. We responded to all affirmatively and corrected/adjusted the text accordingly. Please find responses line-by-line in the attached Word file. 

Reviewer 3 Report

Thank you for this interesting review on the etiopathogenesis of canine cranial cruciate ligament rupture. The manuscript is globally well written but I still recommend some English editing and rephrasing. 

Simple summary:

L16: based on

L18: seem to be

Introduction:

L48: "(primary OA)", I don't understand what you state this here. Please delete

Results:

L162: younger dogs

L167: immune complexes

L176: CLR has not been defined earlier

L178: prior to complete ligament failure but joint instability is present even in a partial tears so I would be very cautious about this sentence. I suggest to modify this part of the sentence.

L180: etiopathology

L195: linked with

L200: correlates

L208: Please rephrase this sentence

L212: Please rephrase this sentence

L217: its

L217: nitric oxide

L221: do you mean treat the pain associated with OA? As far as I know, OA is a incurable disease but you can treat the pain associated to it.

L222: please expand, what do you mean by positive results? less pain? 

L229-230:  it seems that the end of the sentence is lacking

L235: pathogenic factor: at least concomitant to other factors no?

L245: relatively

L258-259: this information is not reported on the drawing and should be to better understand it

L262-264: Please reformulate 

L296: Delete "that"

L296: Please rephrase this sentence

L299: this information is misleading. Please reformulate.

L314: should instead of shall ?

L316:  caudal cruciate and meniscal lesions these are not really typical chronic inflammatory changes but more probably secondary to joint instability!

L333: I suggest: "tends to disappear and become very subtle"

L335: "decreased deambulatory price": at which price? moreover, severe amyotrophy is often present at this stage with a poor functional outcome. I would be very cautious with this sentence and suggest to reformulate or be less affirmative. 

L337: "autostabilisation" : still subnormal! I you were able to get a stifle as stable as before CCLR, I doubt that surgery would still be recommended as the gold standard.  

L341: trochlea

Discussion:

L360: "although rarely rupture": do you have a reference for this assumption?

L396: "autoimmune reaction": do you have a reference?

L401: "have been well documented": you must list the references

L416: "ligament remnants": do you have a reference?

L418: "full limb function": I am not comfortable with this, function is to my opinion still subnormal. Do PVF of untreated dogs have been compared with PVF of operated dogs? I doubt that they would be identical. To my opinion and in my experience, function is still subnormal, at least because of OA.

L437: "satisfactory": excellent outcome is reported for tibial osteotomies! 

L447: not yielded 

To my humble opinion, you should talk a little bit in the discussion about the importance of muscles. Amyotrophy is responsible for the worsening of the instability. Quadriceps has been found to be an CCL agonist (Ramirez et al., 2015) and that can be also kept in mind when attempting non-surgical treatment. 

Author Response

please find attached a Word file with responses to reviewer 2

Round 2

Reviewer 3 Report

Thank you for having considered my suggestions. I have no further recommandation. 

Author Response

Thank you very much for the timely and in-depth review with comments and suggestions we much appreciated and which definitely improved the manuscript